# Reinfection with SARS-CoV-2 and Waning Humoral Immunity: A Case Report

**DOI:** 10.3390/vaccines11010005

**Published:** 2022-12-20

**Authors:** Jason D. Goldman, Kai Wang, Katharina Röltgen, Sandra C. A. Nielsen, Jared C. Roach, Samia N. Naccache, Fan Yang, Oliver F. Wirz, Kathryn E. Yost, Ji-Yeun Lee, Kelly Chun, Terri Wrin, Christos J. Petropoulos, Inyoul Lee, Shannon Fallen, Paula M. Manner, Julie A. Wallick, Heather A. Algren, Kim M. Murray, Jennifer Hadlock, Daniel Chen, Chengzhen L. Dai, Dan Yuan, Yapeng Su, Joshua Jeharajah, William R. Berrington, George P. Pappas, Sonam T. Nyatsatsang, Alexander L. Greninger, Ansuman T. Satpathy, John S. Pauk, Scott D. Boyd, James R. Heath

**Affiliations:** 1Division of Infectious Diseases, Swedish Medical Center, Seattle, WA 98122, USA; 2Providence St. Joseph Health, Renton, WA 98057, USA; 3Division of Allergy and Infectious Diseases, University of Washington, Seattle, WA 98195, USA; 4Institute for Systems Biology, Seattle, WA 98103, USA; 5Department of Pathology, Stanford University, Stanford, CA 94305, USA; 6Department of Microbiology, LabCorp, Seattle, WA 98104, USA; 7LabCorp Esoterix, Calabasas, CA 91301, USA; 8Monogram Biosciences, South San Francisco, CA 94080, USA; 9Swedish Center for Research and Innovation, Swedish Medical Center, Seattle, WA 98104, USA; 10Division of Infectious Diseases, Polyclinic, Seattle, WA 98104, USA; 11Division of Pulmonology and Critical Care Medicine, Swedish Medical Center, Seattle, WA 98104, USA; 12Department of Laboratory Medicine and Pathology, University of Washington School of Medicine, Seattle, WA 98109, USA; 13Vaccine and Infectious Disease Division, Fred Hutch, Seattle, DC 98109, USA; 14Sean N. Parker Center for Allergy and Asthma Research, Stanford, CA 94304, USA

**Keywords:** SARS-CoV-2, COVID-19, reinfection, humoral immunity

## Abstract

Recovery from COVID-19 is associated with production of anti-SARS-CoV-2 antibodies, but it is uncertain whether these confer immunity. We describe viral RNA shedding duration in hospitalized patients and identify patients with recurrent shedding. We sequenced viruses from two distinct episodes of symptomatic COVID-19 separated by 144 days in a single patient, to conclusively describe reinfection with a different strain harboring the spike variant D614G. This case of reinfection was one of the first cases of reinfection reported in 2020. With antibody, B cell and T cell analytics, we show correlates of adaptive immunity at reinfection, including a differential response in neutralizing antibodies to a D614G pseudovirus. Finally, we discuss implications for vaccine programs and begin to define benchmarks for protection against reinfection from SARS-CoV-2.

## 1. Introduction

The risk of reinfection with SARS-CoV-2 after primary infection had not been consistently demonstrated until late in 2020 [1]. Multiple reports documented prolonged viral RNA shedding [2], though virus is not likely to be transmissible after 10 days [3,4], or possibly up to 20 days in immunocompromised patients [5]. These data suggested prolonged shedding of viral remnants, as opposed to ongoing shedding of replication-competent virus. A large case series from the Korean CDC [6] found lack of transmission events from symptomatic patients repeatedly positive for SARS-CoV-2 after negative testing. Most case reports did not distinguish between prolonged shedding and reinfection [7,8,9]. Without viral sequencing analysis, one could not exclude the possibility that prolonged shedding in some patients might actually be reinfection. In August 2020, reports from Hong Kong and Nevada described reinfection 5 and 2 months after primary infection, respectively [10,11]. In this case, report, we describe a third case of reinfection discovered contemporaneously in the summer of 2020.

After SARS-CoV-2 infection, most persons develop anti-SARS-CoV-2 antibody responses characterized by rising IgG, IgM and IgA to viral spike, receptor binding domain (RBD) or nucleocapsid (N) antigens [12]. By 4 weeks after symptoms onset, IgM and IgA decline substantially, as does IgG in patients with mild or asymptomatic infections, while IgG persists at higher levels after severe COVID-19 illness [13]. Evidence suggests that SARS-CoV-2-specific antibodies can be protective, as indicated by the lack of infection in those with pre-existing neutralizing antibodies (nAb) in a recent high attack rate outbreak aboard a fishing vessel [14]. Convalescent plasma programs are based on the assumption that humoral immunity will aid in the response to SARS-CoV-2 [15], as are vaccine programs aiming to provide durable herd immunity [16]. However, correlates of immunity from reinfection have not been established due to the few documented reinfections, and the limited immunological studies in the reports of reinfection [10,11]. Here, we use whole viral genome sequencing to define a reinfection case. We then present antibody, B cell and T cell analyses to evaluate the patient’s lack of immunity against a new SARS-CoV-2 strain.

## 2. Materials and Methods

### 2.1. Patient Population & PCR Testing

“Re-positivity” was defined as repeat SARS-CoV-2 PCR positive after negative testing in patients with initially PCR-confirmed COVID-19. To understand the duration of shedding and phenotypes of re-positivity, we analyzed a database of all SARS-CoV-2 PCR testing for patients with nasopharyngeal samples sent from the emergency departments or hospitals of Swedish Health System in Seattle, WA, USA. Semi-quantitative real time polymerase chain reaction (RT-PCR) testing reported as cycle thresholds (Ct) were performed on the Xpert Xpress SARS-CoV-2 test on the GeneXpert Infinity (Cepheid, Sunnyvale, CA, USA). Hospital policies discouraged unnecessary testing, but the decision to test was left to individual providers. Retesting was often requested by congregate living facilities prior to receiving patients following hospitalization. Discontinuation of transmission-based isolation synchronized with the CDC interim guidance [17]. Descriptive statistics were performed on population shedding dynamics.

### 2.2. Virologic and Immunologic Analyses

Viral sequencing in March was performed via rapid metagenomic next-generation sequencing (NGS) [18], and in July was modified from the multiplexed PCR amplicon NGS method using the ARTIC V3 primers [19]. SARS-CoV-2 clade designations and phylogenetic analyses were produced using NextStrain [20]; sequence acknowledgements are provided in Appendix A. Serological testing used enzyme-linked immunosorbent assay (ELISA) for anti-spike, anti-RBD, and anti-N IgG, IgM and IgA antibodies, as well as a functional assay for antibodies that block binding of RBD to an ACE2 fusion protein [13]. Functional nAb were measured with a cell-culture based assay using pseudoviruses containing either the D614 or the G614 epitopes in spike [21]. Immunoglobulin heavy chain (IGH) genes expressed by peripheral blood B cells were sequenced with amplicon libraries produced for each isotype [22], and paired IGH and light chain sequences were determined with single B cell transcriptome analysis [22]. T cell phenotyping was performed by single cell CITE-seq (10x Genomics, Pleasanton, CA, USA) with dimensional reduction analysis [23]. Plasma cytokine levels were measured with proximity extension assay (Olink Proteomics, Uppsala, Sweden) [23]. All assays are described in depth (Appendix A).

## 3. Results

### 3.1. Population Sampling

Between 1 March and 12 August 2020, 11,622 patients were tested for SARS-CoV-2 by RT-PCR (Appendix A). Of these, 643 patients had at least one positive test (5.5% positivity) and 176 patients had at least two positive samples. Time from first positive to last positive was determined as the shedding duration (Figure 1A). The median (interquartile range) shedding duration was 12.1 (6.4, 24.7) days, with a positively skewed distribution (kurtosis = 10.7). Shedding was <59 days in 95% of patients and was >75 days in only two patients. Re-positivity was observed in 43 patients (Figure 1B) with patterns suggesting: (1) inadequate sampling technique, (2) assay limitations with the Ct result hovering at the limit of detection, (3) prolonged shedding, potentially combined with either of the former, or (4) reinfection. The patient with the longest duration between negative RT-PCR and re-positive was enrolled in an observational study to distinguish between these possibilities.

### 3.2. Case Study

InCoV139 is a sexagenarian (age between 60–69), who resides in a skilled nursing facility (SNF) and has a history of severe emphysema (FEV1 34% predicted) on home oxygen, and hypertension. In late February, the patient visited the emergency room visit for syncope and was diagnosed with exacerbation of chronic obstructive pulmonary disease. Three days later, the patient returned with recurrent syncope, but now also had symptoms including fever, chills, productive cough, dyspnea and chest pain and was admitted to the hospital. At the SNF, the patient reported exposure to a SNF employee who was recently returned from the Philippines with a respiratory infection. Auscultation revealed diffuse wheezing and dullness at the left base and chest X-ray showed hyperinflation and bibasilar infiltrates (Appendix A). Unstable atrial fibrillation ensued and the patient was treated with cardioversion and anticoagulation. On hospital day 6, the patient tested positive for SARS-CoV-2 by rt-PCR, confirming the diagnosis of severe COVID-19 pneumonia. The patient received treatment with supportive care consisting of supplemental oxygen, steroids, and multimodal inhaled therapies for chronic obstructive pulmonary disease. The patient returned to the SNF after testing negative on days 41 and 43 of hospitalization (43 and 45 days after symptoms onset, which was retrospectively determined as the date of first syncope).

After the hospitalization in March for severe COVID-19 pneumonia, InCoV139 remained isolated from family and visitors, interacting only with SNF residents and staff. After moving to a different facility, the patient described exposure to residents at the new facility who were coughing. On day 140 after the first positive SARS-CoV-2 PCR, the patient returned to the ER with dyspnea, reporting 2 weeks of dry cough and weakness. SARS-CoV-2 PCR was repeatedly positive on days 1 and 6 of re-hospitalization (day 14 and day 19 after reinfection date of symptoms onset). Compared to admission in March, the patient was less severely ill in July, by physiologic, laboratory and radiographic parameters, with higher Ct values (Table 1, Appendix A). Status returned to baseline after treatment with remdesivir and dexamethasone. The complete SARS-CoV-2 RT-PCR testing history is given (Appendix A), with variability in testing location in March indicative of the rapid evolution of clinical care processes and limited availability of SARS-CoV-2 diagnostic RT-PCR testing.

### 3.3. Viral Sequencing and Phylogenetic Analysis

Comparison of InCoV139 sequences from March and July revealed 10 high confidence single nucleotide variants (SNVs) of which 5 type the March sequence to clade 19B, and 5 type the July sequence to clade 20A. The InCoV139-March sequence (Genbank: MT252824) shares the canonical mutations (C8782T and T28144C) which define clade 19B and distinguish it from the original clade 19A, of the Wuhan-Hu-1 reference strain (Genbank: NC_045512.2). InCoV139-March additionally shares C18060T with the first US case WA1 (Genbank: MN985325), which was circulating in Asia and introduced via a traveler returning from Wuhan, China to the Puget Sound area north of Seattle in mid-January [23]. InCoV139-March diverges from WA1 by at least 2 other mutations suggesting evolution via community spread in the ensuing 7 weeks from diagnosis of WA1 to diagnosis of InCoV139-March. Conspicuously, the July sequence (InCoV139-July) harbors none of the canonical mutations defining clade 19B and instead shares the canonical mutations defining clade 20A (C3037T, C14408T and A23403G), one canonical mutation of clade 20C (G25563T), as well as one other 20A mutation. Importantly, present in InCoV139-July (but not in InCoV139-March) is the A23403G mutation, which confers the D614G amino acid change in spike protein and defines the SARS-CoV-2 strain with greater replicative fitness, introduced separately to the US East Coast via Europe [18]. As indicated in the phylogeny (Figure 2), sequence differences (Appendix A) clearly define 2 genetically distinct viruses which evolved separately from a common ancestor in early divergent events.

### 3.4. Anti-SARS-CoV-2 Antibody Response

Plasma samples from InCoV139 in July were measured for anti-SARS-CoV-2 antibodies (Figure 3). IgG antibodies against RBD, spike and nucleocapsid were detected, with low optical density compared to positive control [13] and showed a decreasing trend from day 14 to 42 after reinfection symptoms onset. IgM was weakly positive to spike, but undetectable to RBD and nucleocapsid. IgA specific for spike and nucleocapsid, but not RBD, was detected at low levels on day 14 to 21. Anti-spike and anti-RBD IgA showed a surprising increase by day 42, confirmed in replicate and titration experiments (Appendix A). IgG subclass analysis revealed that the patient’s RBD-specific IgG response consisted of low levels of IgG3, without detectable IgG1, despite having both IgG1 and IgG3 specific for spike and nucleocapsid proteins with decreasing trend (Appendix A). Antibodies blocking ACE2-RBD binding were undetectable at day 14, suggesting a lack of potentially protective antibodies, and increased by day 42 (Figure 3). At day 14 and 42, nAb titers (IC50) were 1:260 and 1:382 against D614 (Wuhan-Hu-1) pseudovirus, and were 1:449 and 1:1168 against a mutated D614G pseudovirus, showing differential increase of nAb to D614G pseudovirus compared to the Wuhan-Hu-1 strain (Figure 3D and Appendix A).

### 3.5. Antibody and B-Cell Receptor Repertoires

B cells were evaluated in peripheral blood at day 14 and 18 after reinfection by NGS of IGH genes of all isotypes (Figure 4A). Healthy human peripheral blood shows a predominance of naïve B cells expressing IgM and IgD without somatic hypermutation, and memory B cells with mutated class-switched antibodies. In contrast, the acute response to primary SARS-CoV-2 infection features large polyclonal expansions of recently class-switched, low somatic hypermutation B cells expressing IgG subclasses and, to a lesser degree, IgA subclasses [12], as shown in longitudinal samples from an unrelated patient at day 9 (prior to seroconversion) and day 13 (after seroconversion) after primary infection with SARS-CoV-2. In contrast, clones with low somatic hypermutation did not emerge by day 14 or 18 after reinfection in patient InCoV139 (Figure 4A). Parallel analysis by single-B cell immunoglobulin sequencing revealed elevated frequencies of IgA-expressing B cells, particularly IgA2-expressing cells (Figure 4B).

### 3.6. T Cell Phenotypes and Plasma Proteins

T cells were evaluated at day 14, day 18 and day 21 after reinfection by single cell CITE-seq analysis (Figure 5A) on peripheral blood mononuclear cells (PBMCs). These data from the reinfected patient are compared against similar analytics of PBMCs collected from healthy donors and from a cohort of 26 patients after primary SARS-CoV-2 infection exhibiting varying levels of infection severity. T cell analysis from the reinfected patient demonstrated a consistently unique T cell signature for all three time points: CD8+ T cells had very low levels of naïve-, proliferation-, or exhaustion-related transcripts relative to what is seen in both healthy donors and the COVID-19 primary infection cohort. Further, memory-like markers were upregulated in the reinfected samples. Similarly, CD4+ T cells in the reinfected patient samples exhibited decreased levels of naïve-, proliferation- and exhaustion- related transcripts. Interestingly, the day 21 sample of the reinfected individual displayed the highest levels of Tfh marker CXCR5. This may be associated with the observed increase in the antibody neutralization ability of the reinfected individual on day 42, since Tfh cells are critical for facilitating the maturation of B cells for development of humoral immunity.

In addition to T cells, plasma protein profiles in the reinfected patient also showed interesting patterns (Figure 5B). In the primary infection COVID-19 patient cohort, several plasma proteins increase with COVID-19 disease severity, including IL-6, IL-8, IL-10, IL-4, and VEGF. However, in the reinfected patient, the levels of these proteins were not significantly different from what is observed in the healthy controls, correlating with the observed mild clinical phenotype at the time of reinfection.

## 4. Discussion

We present a case of SARS-CoV-2 reinfection and perform extensive characterizations of antibody and B cell responses. At the time of the case in summer 2020, our data provided a benchmark for understanding the correlates of humoral immunity required to prevent reinfection. Understanding such correlates can aid in planning public health measures as some persons are likely to be at risk for reinfection due to waning antibody-mediated immunity. Understanding protective levels of total anti-spike antibodies and nAb are important for vaccine development.

Molecular evidence for reinfection in our patient is strong. At initial infection during the early outbreak in Seattle, sequences of community circulating viruses had low diversity, and were derived from a founder virus introduced to the US some 7 weeks earlier [24]. By the time of the reinfection, the spike variant D614G from Europe had taken over as the predominant circulating strain [25]. The time course of InCoV139’s two infections overlaps with the transition in Seattle to the newer D614G strain [26], supporting reinfection as opposed to intra-host evolution.

The case patient had anti-SARS-CoV-2 IgG antibodies in the first weeks after reinfection, but notably, levels of anti-RBD IgG were relatively low, with no evidence of blocking antibodies to the RBD-ACE2 complex. ACE2 blocking increased only slightly by day 42, likely due to IgA antibodies, suggesting an impairment in the humoral response to reinfection. In the B cell receptor repertoire, new class-switched clones with low somatic hypermutation were not prominent by day 18 or even by day 42 after reinfection, in contrast to B cell responses seen in primary infection of patients. T cell phenotyping and plasma proteomics suggests a mild response of CD8+ and CD4+ cells with little in the way of inflammation. CD4+ Tfh transcripts did increase by day 42 consistent with this cells role in facilitating the maturation of B cells and humoral immunity. While we do not know the nAb titers immediately at the time of reinfection, by day 14 after reinfection, nAb levels were comparable to those observed after boosted vaccination [21]. By day 42 nAb response showed a 1.5-fold increase to Wuhan-Hu-1 pseudovirus, and a 2.6-fold increase to D614G pseudovirus. Taken together, these findings suggest that poorly developed or waned antibodies against the D614 virus formed after primary infection in March 2020 were not protective against reinfection with the D614G spike variant acquired in July 2020. These results have important implications for the success of vaccine programs based on the Wuhan-Hu-1 strain.

Fortunately for our patient, the reinfection was milder than was the primary infection, in contrast to the Nevada case [11]. This case report provides an initial in-depth assessment of adaptive immune responses including T cell, B cell and antibody-mediated immunity during reinfection. The humoral immunity in this patient was clearly insufficient to prevent reinfection, and therefore provided a starting point for the development of benchmarks for antibody responses associated with protective immunity. Larger case series of reinfection patients or follow-up experience after vaccination studies will be needed to more thoroughly evaluate correlates of immune protection against SARS-CoV-2.

## 5. Conclusions

This paper was drafted in August 2020 and is largely unmodified from the manuscript we finalized as a preprint in September 2020 [27]. In addition to our results on humoral immunity, this publication represents an important snapshot or time capsule of the state of scientific response early in the COVID19 pandemic. Notably, at that time, there was tremendous skepticism of the possibility of reinfection with SARS-CoV-2 (e.g., [28]). This skepticism persisted despite periodicity (waves) in infection counts after outbreaks that should have resulted in herd immunity if textbook assumptions were valid [29]. The first outbreak in Manaus was one such example [30]. Waning immunity and reinfection are now recognized to be more important factors for COVID19 epidemiology than was originally anticipated. Understanding reinfection—causes, mechanisms, and consequences—will be important for future public health responses to emerging pathogens. As of late 2022, SARS-CoV-2 reinfection has been increasingly studied and reviewed (e.g., [31,32]). As the current COVID19 pandemic moves towards endemicity, near constant infection rates in some populations may indicate a herd-immunity steady state in which waning immunity in previously immune individuals balances renewed immunity due to vaccinations and/or infections. The length of immunity conveyed by vaccination and infection, which will vary between individuals and over time, will in large part determine the nature and public health impact of endemic COVID19.

## Figures and Tables

**Figure 1 vaccines-11-00005-f001:**
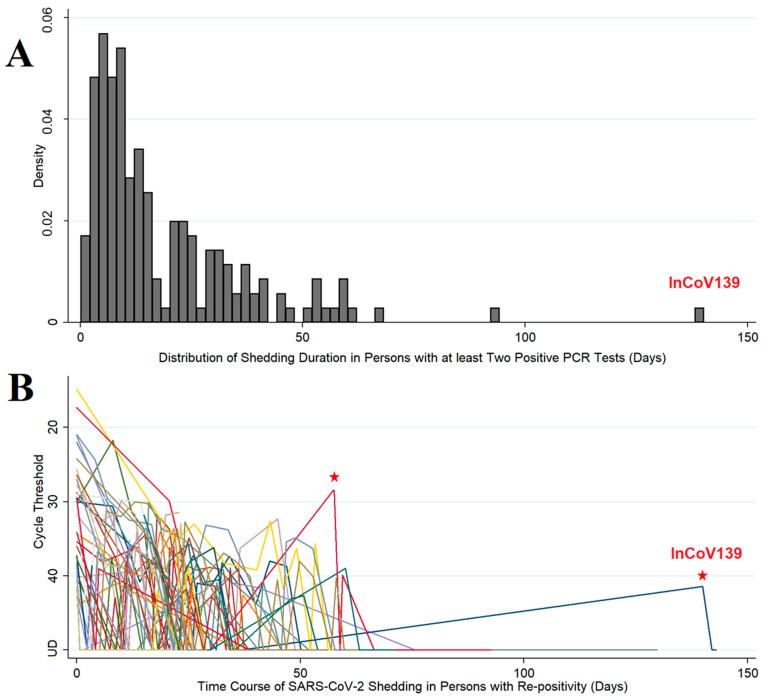
Population viral RNA shedding from patients with COVID-19. Panel (**A**): Distribution of shedding duration in patients who had at least 2 positive SARS-CoV-2 PCR tests. The shedding duration was calculated as the time from first positive sample to last positive sample. In the histogram (n = 176), the proportion of patients is plotted as density on the *y*-axis and shedding duration (in days) is on the *x*-axis. Panel (**B**): Time course of SARS-CoV-2 shedding in patients (n = 43) who had “re-positive” pattern (repeat SARS-CoV-2 PCR positive after negative testing in patients with initially PCR-confirmed COVID-19, i.e., a positive-negative-positive pattern). In the spaghetti plot, semi-quantitative real-time PCR expressed in cycle thresholds (Ct) is plotted on the *y*-axis and time course in days from first positive to last positive is on the *x*-axis. Ct is the average result of E & N2 genes except where one target was undetectable and then Ct was set to value of single positive target. Ct range: 14.9–44.0. Negative (undetectable) results are set to Ct = 50 for purposes of display. UD = undetectable. Red stars mark possible reinfections due to low CT value at re-positive, or long duration since last positive PCR, respectively.

**Figure 2 vaccines-11-00005-f002:**
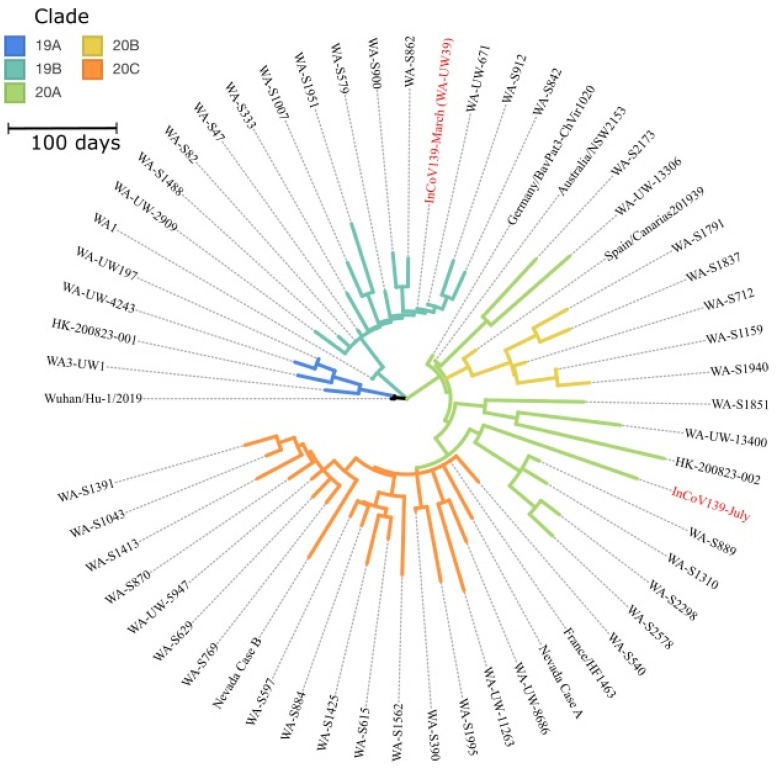
Phylogram of SARS-CoV-2 Isolates in Washington State. A phylogeny of SARS-CoV-2 in Washington State demonstrates that the primary and reinfection strains are distinct. The figure also includes contextual sequences illustrating clade roots and the pairs of reinfection cases from Hong Kong and Nevada. The pair of reinfection samples from InCoV139 (red) type to clade 19B from the primary infection in March and to clade 20A from the reinfection in July. The initial Hong Kong sample was in Clade 19A and the reinfection sample in 20A; both Nevada samples were in Clade 20C. Sequences labels follow abbreviated GISAID nomenclature, and further sequence information is provided (Appendix A). An interactive version of this tree is included as Appendix A.

**Figure 3 vaccines-11-00005-f003:**
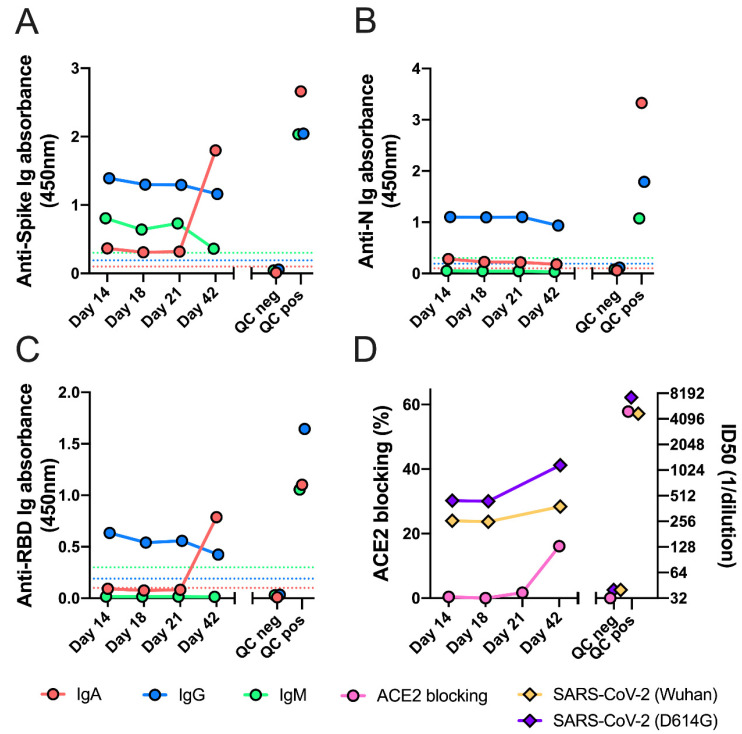
Anti-SARS-CoV-2 serologies and neutralizing antibodies. Plasma samples were analyzed by ELISA at a 1:100 dilution for the presence of IgG, IgA and IgM antibodies binding to the SARS-CoV-2 spike (Panel (**A**)), nucleocapsid (Panel (**B**)), and RBD (Panel (**C**)) antigens. Panel (**D**) shows the results of testing for antibodies that block the binding of ACE2 to RBD, carried out with a 1:10 dilution of plasma (left *y*-axis). Pseudovirus neutralizing antibodies were detected with in vitro cell culture assay with D614 (Wuhan-Hu-1) pseudovirus and D614G pseudovirus (right *y*-axis). For all panels, time on the *x*-axis indicates days after symptom onset during SARS-CoV-2 reinfection. Plasma pools from SARS-CoV-2 pre-pandemic healthy blood donors and from primary infection COVID-19 patients were used as negative and positive quality control (QC), respectively. The dotted lines are the cutoffs value for a positive result for each assay, determined as described in the Appendix A. All samples were tested in duplicate wells; mean OD values are shown. Results are shown from one of two replicate experiments carried out on different days.

**Figure 4 vaccines-11-00005-f004:**
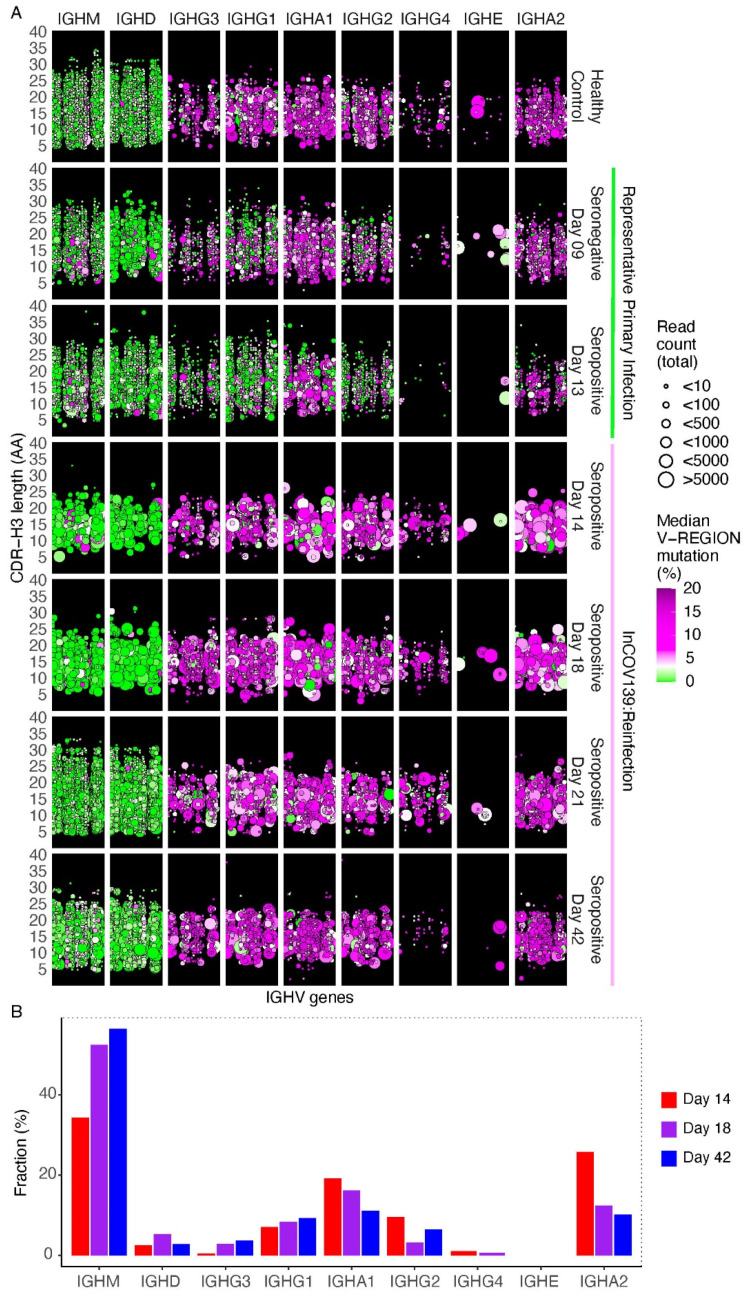
B cell repertoire responses. Panel (**A**): Peripheral blood B cell IGH gene repertoires from peripheral blood mononuclear cell RNA. Three individuals were sampled: a representative healthy control (top row); a patient with representative primary SARS-CoV-2 infection at day 9 and 13 post-onset of symptoms (highlighted by green bar); and reinfected patient at day 14, 18 and 42 (pending) post-onset of symptoms (highlighted by pink bar). Serostatus and days post symptoms onset are given on the right *y*-axis. Columns indicate the class of each IGH sequence with the IGHV gene indicated on the *x*-axis. The left *y*-axis indicates CDR-H3 length in amino acids (AA). Dots indicate B cell clonal lineages. Point color indicates median IGHV somatic hypermutation frequency for each clone, and point size indicates the number of reads in the clone. Points are jittered to decrease over-plotting. Panel (**B**): The bar plot summarizes single-B cell transcriptome data indicating the antibody isotype expressed by B cells in the reinfected patient’s blood, plotted as the frequency of usage of each isotype.

**Figure 5 vaccines-11-00005-f005:**
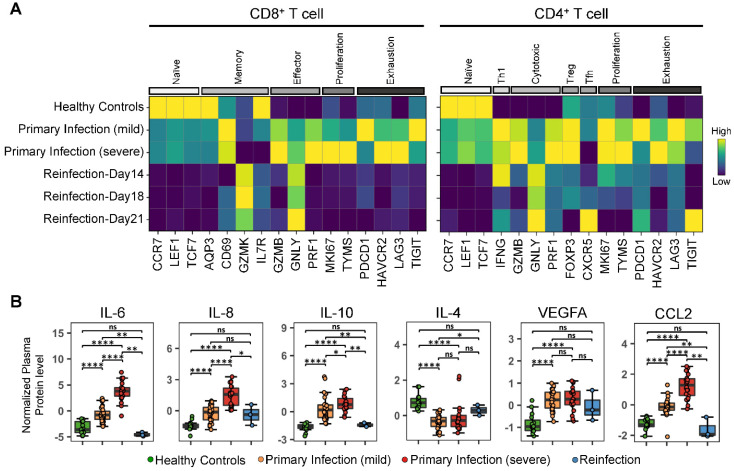
T cell phenotypes and plasma proteins. T cell phenotypes are compared between the reinfection patient (InCoV139), 26 representative primary infection patients sampled at 2 time points with mild infection (n = 29 samples) or severe infection (n = 21 samples), as well as 6 healthy controls sampled prior to the SARS-CoV-2 pandemic (n = 6 samples). For the primary infection cohort, severity at the time of sample draw was determined by applying the World Health Organization (WHO) ordinal scale score (see Appendix A) as follows: mild = WHO 1–4, severe = WHO 5–7. Panel (**A**): Heatmaps showing the normalized expression of select phenotype-specific representative gene expression in CD8+ and CD4+ T cells for healthy controls (top row), primary infection patients (rows 2–3) and the reinfection patient at different time points (bottom 3 rows). The heatmap color describes relative abundance. Panel (**B**): Box plots of select plasma protein levels from the same healthy, COVID-19 patients and the reinfected individual (all 3 time points). Plasma proteins are chosen as representative markers of inflammation (IL-6), stimulatory (IL-8), regulatory (IL-10, IL-4), and chemoattractive (CCL2) cytokines. We also included VEGF level for comparison given the key role of this marker in hypoxia. Significance is indicated by: (ns = not significant, * *p* < 0.05, ** *p* < 0.01, **** *p* < 0.0001).

**Table 1 vaccines-11-00005-t001:** Clinical Parameters at Peak Illness for COVID-19 Episodes.

Parameter:	Primary Infection (March) *	Reinfection (July) *
Vital Signs:		
Temperature (°C)	38.4	37.0
Heart Rate (/min)	101	86
Blood pressure (mmHg)	156/96	143/93
Respiratory Rate (/min)	20	19
SpO2 (%) on supplemental O_2_ rate	93% on 6 L/min	94–97% on 3 L/min
BMI (kg/m^2^)	18.7	20.4
Laboratory:		
Total white blood count (cells/µL)	16,200	6700
Absolute neutrophil count (cells/µL)	12,960	2010
Absolute lymphocyte count (cells/µL)	1600	600
Hematocrit (%)	39.6%	42.8%
Platelet count (cells/µL)	290,000	240,000
D-dimer (≤0.49 µg/mL) **	N/A	0.47
Creatinine (mg/dL)	1.01	1.07
Procalcitonin (≤0.25 ng/mL) **	0.15	0.08
C-reactive protein (≤5 mg/L) **	N/A	<3.0
SARS-CoV-2 rt-PCR CT (target 1) †	22.8 (E)	43.3 (E)
SARS-CoV-2 rt-PCR CT (target 2) †	26.5 (RdRp)	39.6 (N2)

* Peak day of illness for each COVID-19 episode is given. For primary infection, peak illness occurred in the 1st hospitalization (March) on hospital day #5. For reinfection, peak illness occurred in the 2nd hospitalization (July) on hospital day #1. ** Normal ranges given as indicated. † SARS-CoV-2 qualitative polymerase chain reaction (rt-PCR) cycle threshold (CT) was based on the WHO assay in March (UW Virology) and the Cepheid Infinity in July (LabCorp Seattle).

## Data Availability

The full data is available upon request. Code used to generate analyses is available upon request. Viral sequences are available through GenBank.

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
