# Peer review of "Reinfection with SARS-CoV-2 and Waning Humoral Immunity: A Case Report"

_vaccines, 2022, doi:10.3390/vaccines11010005_

Round 1

Reviewer 1 Report

The manuscript of Goldman et al. describes an in-depth study of a single patient apparently re-infected with a different SARS-CoV2 strain after recovering from the first infection. The study is scientifically sound and well-presented but has two or even three major issues that should be resolved prior to publication: 1) Supplementary Figures and Methods are unavailable on the journal website, the file with the same exact Figures as within the manuscript is uploaded instead, therefore full data review is impossible; 2) Authors have to address and explain, at least in the Discussion (in addition to Conclusions), why this manuscript is submitted more than two years after the fact and to correct statements made in Discussion with respect to 2+ years of SARS-CoV research post-dating their results (otherwise Discussion looks really obsolete and not at all helpful). Additionally, using the words "Failure of Humoral Immunity" in the title of the manuscript (notably, not in the Abstract) is too strong a statement for this reviewer, it should be definitely amended, especially in light of the patient having much weaker disease on the re-exposure and also presenting certain (albeit not all) immune correlates of protection.

Minor comments relate to a few typos: 1) The name of 'Shannon Fallen and BS' in the authors' list should probably be just 'Shannon Fallen'; 2) The reference 20 (line 89), if this is actually it, is not properly formatted; 3) 'Plasma cytokines levels' (line 94) should be 'Plasma cytokine levels'; 4) The word 'patient' is missing from the sentence in lines 133-134 (should be 'and THE PATIENT was treated').

Reviewer 2 Report

COVID-19 continues to be an evolving disease, with the emergence of a number of SARS-CoV-2 genetic variants impacting the efficacy of currently approved COVID-19 vaccines. Research is needed to understand the host immune response to viral variants, the kinetics of neutralizing antibodies, and correlates of immunity associated with SARS-CoV-2 reinfection. In this manuscript, Goldman et al. provide a very thorough genetic and immunological investigation of a documented case of reinfection with SARS-CoV-2. The authors used whole-genome sequencing to compare the viruses of the initial and reinfection cases and analyzed antibody production, B cells, and T cell phenotypes to provide insight into correlates of adaptive immunity from reinfection. While this study has the limitations inherent in case reports (i.e., a very small sample size and in this case, one patient), the authors provide valuable information on the antibody, B cell, and T cell analytics associated with this case of SARS-CoV-2 reinfection. This case report is very interesting, exceedingly well-written, and provides important information on the immunological characteristics of reinfection. I could not find any major or minor issues that need to be addressed by the authors prior to publication.
